# Endoscopic Intragastric Injection of Botulinum Toxin A in Obese Patients Accelerates Weight Loss after Bariatric Surgery: Follow-Up of a Randomised Controlled Trial (IntraTox Study)

**DOI:** 10.3390/jcm11082126

**Published:** 2022-04-11

**Authors:** Francisco José Sánchez-Torralvo, Luis Vázquez-Pedreño, Montserrat Gonzalo-Marín, María José Tapia, Fuensanta Lima, Eduardo García-Fuentes, Pilar García, Javier Moreno-Ruiz, Alberto Rodríguez-Cañete, Sergio Valdés, Gabriel Olveira

**Affiliations:** 1Unidad de Gestión Clínica de Endocrinología y Nutrición, Hospital Regional Universitario de Málaga, 29010 Malaga, Spain; montserratgonzalomarin@gmail.com (M.G.-M.); mjtapiague@gmail.com (M.J.T.); santi.lima@hotmail.com (F.L.); sergio.valdes@hotmail.es (S.V.); gabrielm.olveira.sspa@juntadeandalucia.es (G.O.); 2Departamento de Medicina y Dermatología, Facultad de Medicina, University of Malaga, 29010 Malaga, Spain; 3Instituto de Investigación Biomédica de Málaga-IBIMA, 29010 Malaga, Spain; edugf1@gmail.com; 4Unidad de Endoscopias, Unidad de Gestión Clínica de Aparato Digestivo, Hospital Regional Universitario de Málaga, 29010 Malaga, Spain; luichivazquez@hotmail.com (L.V.-P.); pilar007s@gmail.com (P.G.); 5Centro de Investigación Biomédica en Red de Diabetes y Enfermedades Metabólicas Asociadas (CIBERDEM), Instituto de Salud Carlos III, 28029 Madrid, Spain; 6Unidad de Gestión Clínica de Aparato Digestivo, Hospital Universitario Virgen de la Victoria, 29010 Malaga, Spain; 7Centro de Investigación Biomédica en Red de Enfermedades Hepáticas y Digestivas (CIBEREHD), Instituto de Salud Carlos III, 28029 Madrid, Spain; 8Unidad de Gestión Clínica de Cirugía General y Digestiva, Hospital Regional Universitario de Málaga, 29010 Malaga, Spain; javier.morenoruiz@gmail.com (J.M.-R.); arodriguezcane@hotmail.com (A.R.-C.)

**Keywords:** obesity, botulinum toxin, weight loss, endoscopy, bariatric surgery

## Abstract

Background: Intragastric injection of botulinum toxin A (BT-A) has been shown to be effective for weight loss up to six months after administration, according to previous studies. Our objective was to determine, in patients on bariatric surgery waiting lists, the effect of BT-A on weight loss in the pre- and postoperative period and to analyse if there are different responses based on Body Mass Index (BMI). Methods: We performed a follow-up analysis of the IntraTox study, which included 46 patients on bariatric surgery waiting lists in a single-centre, randomised, double-blind, placebo-controlled clinical trial. The treatment group received intragastric BT-A, whereas the control group received physiological saline solution. The one-time procedure was performed at the time of diagnostic endoscopy 7–8 months before surgery. Weight loss was evaluated at admission and after 4 and 12 weeks from the bariatric surgery. Our analysis was stratified by BMI at randomisation. Results: weight loss percentage on the day of surgery, with respect to the initial visit, was −4.5 ± 3.9% for the control group vs. −7.6 ± 4.2%, for the treatment group (*p* = 0.013). Weight loss percentage tended to remain greater in the treatment group one month after the intervention (−12.7 ± 4.7% vs. −15.2 ± 4.6%, *p* = 0.07) and become similar three months after (−21.6 ± 4.7% vs. −21.6 ± 4.6%). After stratifying by BMI, only patients with BMI over 50 kg/m^2^ allocated to the treatment group obtained a greater weight loss at the end of the trial, the day of surgery, and one month after, compared with the placebo group (−4.9 ± 4.9%, −10.8 ± 5.3% and −17.1 ± 3.8% vs. −0.1 ± 2.6%, −4.3 ± 3.2% and −12.8 ± 4.1%, respectively (*p* < 0.05). Conclusions: intragastric injection of BT-A is effective to achieve significant weight loss, especially in extreme obesity. Its use before bariatric surgery enhances perioperative weight loss.

## 1. Introduction

Obesity is a major public health concern worldwide, as it is a major risk factor for cardiovascular, metabolic, and oncological diseases, among many others [1]. The treatment of obesity is, therefore, an emerging and innovative field with a clinical need of increasing interest in public health. Today, the most effective treatment for weight loss in obesity is bariatric surgery, which is superior to lifestyle modifications and even reduces the occurrence of major adverse cardiovascular events (MACEs) in obese patients with cardiovascular diseases [2]. Nevertheless, it entails invasive procedures and has a substantial risk of adverse events, compared with other treatments [3]. On the other hand, endovascular bariatric surgery (EBS) could be considered an appealing alternative treatment for weight management and cardiovascular prevention in morbidly obese patients with high surgical risk [4].

In search of new treatments for obesity, intragastric injection of botulinum toxin A (BT-A) has been evaluated as a treatment for obesity due to its potential ability to produce delayed gastric emptying and earlier satiety, ultimately causing a reduction in intake and weight loss [5].

Different studies have been carried out to assess the effectiveness and safety of this procedure. Most of them were case series or clinical trials, in a reduced number of cases. The reviews and meta-analyses carried out to date showed varied results with respect to weight loss [6,7,8]. In the IntraTox clinical trial, carried out by our group, we found that the use of endoscopic intragastric injection of botulinum toxin A is both effective and safe for weight loss in obese patients awaiting bariatric surgery [9].

Following the emergence of new studies supporting the use of BT-A as a treatment for obesity [9,10,11,12], the hypothesis arises that this method may be more effective in some specific population subgroups. A recent meta-analysis concluded that the benefit of intragastric BT-A injection in the reduction in total weight occurs in patients with baseline BMI greater than 40 kg/m^2^ [8].

In our centre, all patients awaiting bariatric surgery have to undergo a diagnostic endoscopy several months before the surgery. In the two weeks prior to surgery, a very low-calorie diet (VLCDs) is prescribed to patients to maximise weight loss.

To our knowledge, to date, there are no studies evaluating the effect of intragastric injection of BT-A beyond 6 months of follow-up, as well as its effect after bariatric surgery.

We propose that the use of intragastric BT-A could enhance weight loss prior to bariatric surgery and that its effects could continue after the intervention. Therefore, the objective of our study was to assess if the intragastric injection of BT-A in patients awaiting bariatric surgery results in an increased weight loss preoperatively and at 4 and 12 weeks after the intervention, compared with the control group, as well as the existence of a different response according to baseline BMI.

## 2. Materials and Methods

This analysis included all individual data from the double-blind, placebo-controlled IntraTox (EudraCT 2015-004391-29) study [8] and the postoperative follow-up. IntraTox was a single-centre, randomised, double-blind, placebo-controlled clinical trial performed in the Regional University Hospital of Málaga including patients with obesity on bariatric surgery waiting lists during the period from June 2016 to October 2018.

Considering as clinically significant an expected effect of 5% of weight loss, and to obtain a statistical power of 80–90%, a study of 21 to 27 patients per arm was required. Taking into account possible losses to follow-up, it was decided to include 30 patients per study arm.

The trial enrolled 52 patients randomised 1:1 to receive an intragastric injection of BT-A or placebo, and outcomes included weight loss and quality of life.

After the follow-up related to the clinical trial, the patients were monitored before and after the bariatric surgery intervention. Within the treatment group, one of the patients was excluded due to a delay in the surgery date (more than six months from the end of the clinical trial). It should be noted that one of the patients did not undergo surgery due to a weight loss of 22.9 kg (24.7% of weight loss percentage), resulting in a BMI of less than 30 kg/m^2^. Regarding the control group, the data of four patients were not analysed, since two of them refused surgery and another two patients suffered a delay in the surgery date. The flow of patients throughout the entire study is displayed in the flow diagram (Figure 1).

### 2.1. Endoscopy

Patients were randomised 1:1 to receive an intragastric injection of BT-A or placebo. The one-time procedure was performed at the time of diagnostic endoscopy 7–8 months before surgery. For the treatment group, the preparation consisted of a solution containing 200 U of incobotulinumtoxin A (Xeomin^®^; Merz Pharma, Frankfurt am Main, Germany). From this solution, 1 mL was injected into 16 points of the gastric muscular layer with an endoscopic needle. Four punctures were performed in equidistant circles at 3, 5, and 7 cm from the pyloric ring, and at the fundus level. In the control group, 1 mL of physiological saline solution was injected into each point [9].

### 2.2. Patient Follow-Up

A total of six visits were scheduled for each patient in the original trial: baseline (three days before the endoscopy), and follow-up visits at weeks 2, 4, 8, 16, and 24 after the procedure [9]. In this study, an additional follow-up was carried out after the bariatric surgery. Weight measurements were made on the day of the intervention and at 4 and 12 weeks from the surgery.

### 2.3. Statistical Analysis

Data were computerised and processed via a prefixed and validated clinical data management system (IBM SPSS Statistics, Version 22.0. Armonk, NY, USA). A descriptive analysis of all obtained variables was performed. Data were presented as mean values and proportions. The hypothesis contrasting between proportions was performed using a chi-squared test, and between continuous variables, diverse tests were used: non-parametric tests (U Mann–Whitney or Wilcoxon) in case of continuous variables non-adjusting to normality, and Student’s ANOVA/T in case of continuous variables adjusting to normality. In all cases, the null hypothesis was rejected for an alpha ≤ 0.05 for two tails.

### 2.4. Ethics

All subjects signed the informed consent for inclusion before they participated in the study. The study was conducted in accordance with the Declaration of Helsinki, and the protocol was approved by the Ethics Committee of the Regional University Hospital of Málaga (#26112015) and by The Spanish Agency for Medicine and Health Products (EudraCT: 2015-004391-29).

## 3. Results

A total of 46 patients were included: 20 from the control group and 26 from the treatment group. Bariatric surgery consisted of sleeve gastrectomy in 39 cases and gastric bypass in 7 cases. No significant differences were found in the surgical procedure between the study groups (*p* = 0.112).

Of the 46 participants, 40 were female; their mean age was 43.9 ± 9.2 years (range 26–62 years), their mean BMI in the initial visit was 48.6 ± 5.4 kg/m^2^ (range: 36.7–60.8 kg), and their mean weight was 130.5 ± 17.4 kg (range: 97.8–163.5 kg).

There were no baseline differences regarding weight, BMI, other anthropometric characteristics, and comorbidities between groups. These results are shown in Table 1.

Regarding the safety of this procedure, patients showed symptoms of earlier satiety [9], but no relevant side effects were found in our trial either before or after bariatric surgery.

Weight change over time is shown in Figure 2. Weight loss percentage on the day of bariatric surgery, with respect to the initial visit of the IntraTox study, was −4.5 ± 3.9% for the control group vs. −7.6 ± 4.2%, for the treatment group (*p* = 0.013). Weight loss percentage tended to remain greater in the patients in the treatment group one month after the intervention (−12.7 ± 4.7% vs. −15.2 ± 4.6%, *p* = 0.07) and become similar three months after (−21.6 ± 4.7% vs. −21.6 ± 4.6%).

Among the patients from the treatment group, 17 (65.4%) had lost at least 5% of their weight by the day of the surgery; such weight loss was achieved by 7 patients (35%) from the control group (*p* = 0.041).

The results of weight loss percentage after stratifying by BMI are shown in Figure 3. In the control group, 10 patients had a BMI > 50 kg/m^2^, compared with 8 patients in the treatment group. Only patients with BMI over 50 kg/m^2^ allocated to the treatment group obtained a greater weight loss, compared with baseline at the end of the study, on the day of bariatric surgery, and one month after the intervention, compared with the placebo group (−4.9 ± 4.9%, −10.8 ± 5.3%, and −17.1 ± 3.8% vs. −0.1 ± 2.6%, −4.3 ± 3.2%, and −12.8 ± 4.1%, respectively, with significant differences between groups at all times (*p* = 0.015, 0.006 and 0.044, respectively). Patients with BMI below 50 kg/m^2^ allocated to the treatment group reached a significant reduction in body weight percentage with respect to baseline at the end of the study, on the day of bariatric surgery, and one month after the intervention, but no significant differences were found when comparing with the placebo group (−2.9 ± 6.2%, −6.4 ± 3.1%, and −14.5 ± 4.7% vs. −0.7 ± 4%, −4.6 ± 4.7%, and −12.5 ± 5.5%, respectively; *p* = 0.22, 0.31 and 0.73, respectively).

## 4. Discussion

Our study demonstrated that intragastric injection of botulinum toxin results in weight loss in obese patients awaiting bariatric surgery, especially reaching a significant weight loss percentage in patients with a BMI greater than 50 kg/m^2^.

In obese patients, the use of intragastric BT-A is based on the principle that it reduces gastric motility, resulting in earlier satiety and weight loss. In the past two decades, several studies with different designs have been published [13,14,15,16,17,18], showing mixed results [5].

The effectiveness of the technique has always been under scrutiny, even after performing several meta-analyses [6,7,8]. In our study [9], patients in the treatment group exhibited a significantly greater loss of total weight and a higher percentage of weight loss, compared with the control group.

In the IntraTox study, with a relatively similar sample to the largest clinical trial ever conducted [16], patients in the treatment group showed a significantly higher loss of total weight, a higher percentage of weight loss, and a higher percentage of fat loss, compared with those in the control group [8]. In the present follow-up study, the weight reduction effect was retained until at least the day of bariatric surgery. The participants in our study were instructed to follow a hypocaloric diet, and a very low-calorie diet (VLCD) two weeks before the intervention, although no strict follow-up was carried out. Therefore, the different effect produced on weight between groups seems to be due to the toxin.

Furthermore, even though previously published studies showed the procedure to have scarce effects on weight after the first three months [15,16], IntraTox study results highlighted weight loss gradually from week 2 after the endoscopy until 6 months after it [9]. The analysis of our subsequent follow-up confirms these data since the weight loss was maintained until the day of the intervention and was maintained even up to a month after it in the case of patients with a BMI greater than 50 kg/m^2^, with the potential reduction in postoperative complications.

Although the design of our study was carried out in accordance with the scientific evidence based on previous studies at that time [5,6], more evidence has been collected thereafter to optimise the choice of the target group of patients who benefit from intragastric injections of BT-A [8,10,12].

In our study, we found that patients with extreme obesity (BMI greater than 50 kg/m^2^) had a better weight response than patients below that figure. This result is consistent with a recent meta-analysis [8]. In turn, other studies suggest that the use of intragastric BT-A would not be very effective in overweight patients [11]. These findings could justify the use of intragastric BT-A only in patients with extreme obesity.

In the search for target patients who better benefit from the technique, a recent study [12] suggests that the administration of BT-A would only be associated with weight loss in patients with pyloric normal function, but it would not be present in patients with hypotonic pyloric sphincter, which is why sphincter function should be assessed before BT-A administration.

This follow-up analysis has some limitations. It included neither an exhaustive follow-up of the diet nor an evaluation of physical exercise, and we did not assess the pyloric function before the BT-A administration. On the other hand, after stratifying patients by BMI, the analysis may have been underpowered to identify all predictors of treatment response. Results should be interpreted with caution, and further research investigating factors underlying BT-A response is required. The results of this study have limited generalizability to the overall patient population with obesity due to the exclusion criteria of the IntraTox study [9].

As regards strengths, our work is double-blind, includes an adequate number of patients, and administers BT-A according to the recommendations of previous reviews [5,6].

The percentage of weight loss in our sample would not justify the use of this treatment as first-line therapy; nonetheless, it could be a useful tool for weight loss before bariatric surgery if applied in a preoperative endoscopy, especially in patients with a BMI greater than 50 kg/m^2^.

## 5. Conclusions

These data provide some additional information about the target patient who benefits from intragastric injections of BT-A. According to our results, intragastric injection of botulinum toxin A is an effective and safe procedure to achieve a moderate but significant weight loss percentage, especially in patients with a BMI greater than 50 kg/m^2^ prior to bariatric surgery. Its use before bariatric surgery could enhance perioperative weight loss, although further studies are needed to recommend its widespread use.

## Figures and Tables

**Figure 1 jcm-11-02126-f001:**
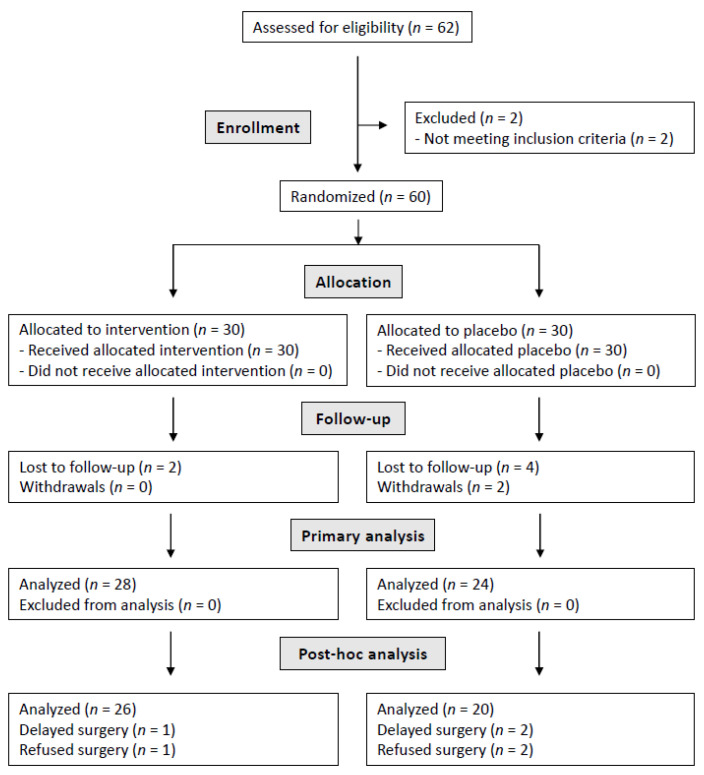
Study flow diagram.

**Figure 2 jcm-11-02126-f002:**
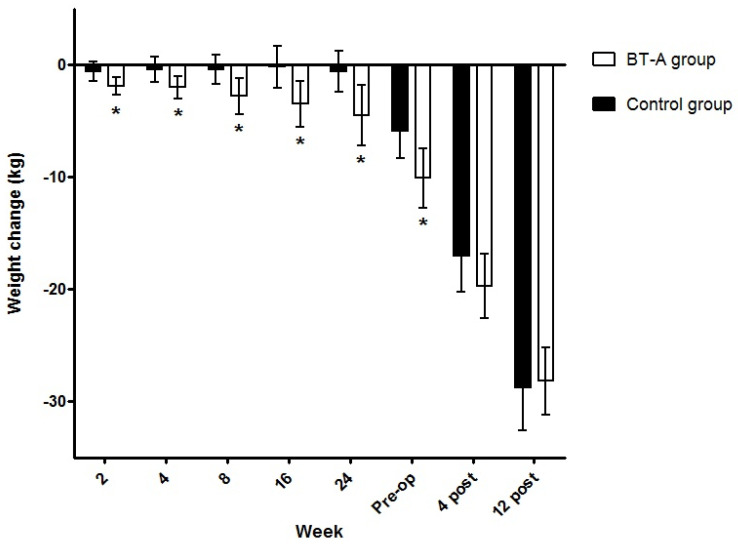
Changes in body weight after intragastric injections. Lines represent mean body weight change over time for each treatment group. * Statistically different means versus control group. Abbreviations: BT-A, botulinum toxin A: pre-op, preoperative; post, postoperative.

**Figure 3 jcm-11-02126-f003:**
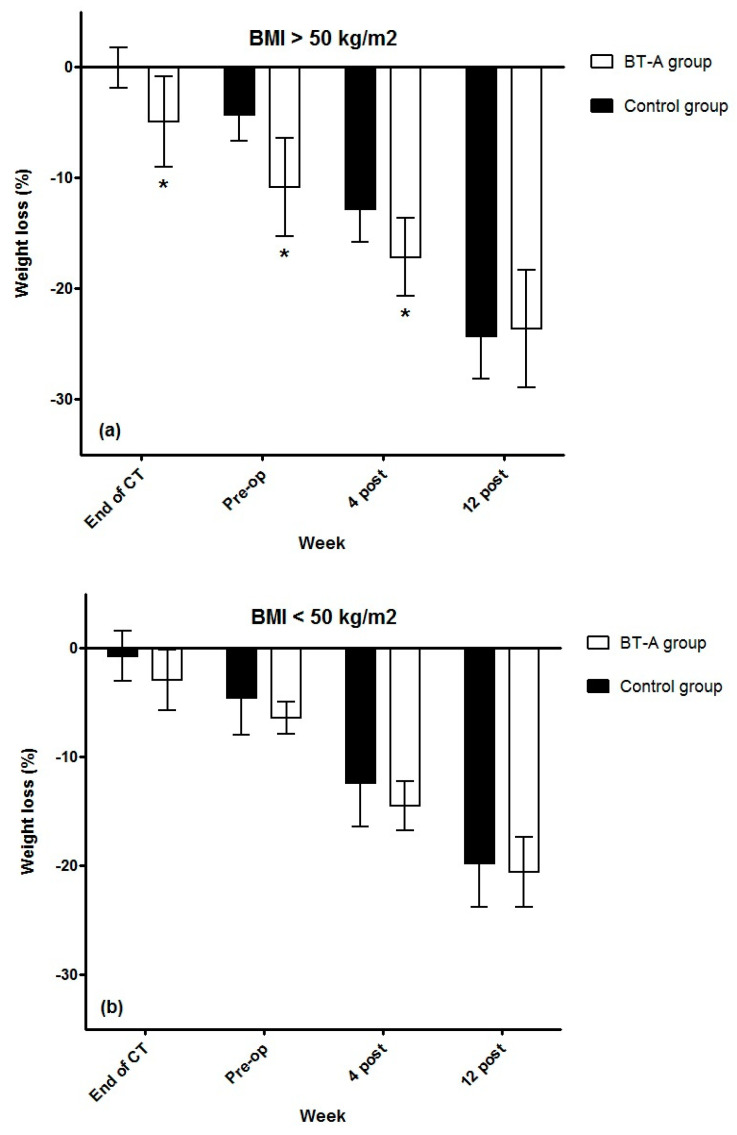
(**a**) Changes in body weight percentage at perioperative period in patients with BMI over 50 kg/m^2^; (**b**) Changes in body weight percentage at perioperative period in patients with BMI below 50 kg/m^2^. * Statistically different means versus control group. Abbreviations: BT-A, botulinum toxin A; CT, clinical trial; pre-op, preoperative; post, postoperative.

**Table 1 jcm-11-02126-t001:** Baseline characteristics: comparison between groups.

	Control Group (*n* = 20)	BT-A Group (*n* = 26)	*p* Value
Age (years) Mean ± SD (min–max)	41.9 ± 7.9 (26–55)	45.6 ± 9.9 (26–62)	0.18
Weight (Kg) Mean ± SD (min–max)	133.7 ± 15.9 (107.3–161.9)	128 ± 18.4 (95.8–163.5)	0.28
BMI (m^2^/Kg) Mean ± SD (min–max)	49.9 ± 4.3 (42.5–56.6)	47.7 ± 6.1 (36.7–60.8)	0.17
Body fat percentage (%BF) Mean ± SD (min–max)	47.8 ± 3 (41.7–52.7)	47.2 ± 4.8 (38.2–55)	0.65
Systolic blood pressure (mmHg) Mean ± SD (min–max)	131.9 ± 14.3 (103–158)	129.5 ± 15.3 (103–157)	0.58
Diastolic blood pressure (mmHg) Mean ± SD (min–max)	78.9 ± 8.7 (65–91)	79.7 ± 10.5 (56–99)	0.77
Gender, female/male (%)	17/3 (85/15%)	18/8 (69/31%)	0.30
Comorbidities (%)	17 (85%)	21 (81%)	0.71
Diabetes mellitus (%)	1 (5%)	6 (19%)	0.21
Hypertension (%)	11 (55%)	9 (35%)	0.23
Dyslipidaemia (%)	4 (20%)	6 (23%)	0.80
OSAS (%)	7 (35%)	10 (38%)	0.81
CVD (%)	3 (15%)	4 (15%)	0.97

Abbreviations: BMI, body mass index; BT-A, botulinum toxin A; SD, standard deviation; OSAS, obstructive sleep apnoea syndrome; CVD, cardiovascular disease.

## Data Availability

Not applicable.

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
