# Peer review of "Endoscopic Intragastric Injection of Botulinum Toxin A in Obese Patients Accelerates Weight Loss after Bariatric Surgery: Follow-Up of a Randomised Controlled Trial (IntraTox Study)"

_jcm, 2022, doi:10.3390/jcm11082126_

Round 1
Reviewer 1 Report
I read with interest the study entitled "Endoscopic intragastric injection of botulinum toxin A in obese patients accelerates weight loss after bariatric surgery: follow-up of a randomized controlled trial (IntraTox study)".
The study is interesting but there are some major critical issues that need to be clarified.
1. How was the sample size calculation done? The authors need to explain this point.
2. What type of bariatric surgeries were performed? Please report this information.
3. Was randomization also performed for the different type of surgery?
4. Fig 2 in the result is not easily understood. The weeks counted in the horizontal line are from 2 to 24 and perhaps it could be an error.
5. Clarify the number of patients above 50 kg/m2 in the 2 groups.
6. The statement on line 141-146 in the results paragraph: "Only patients with BMI greater than 50 kg/m2 assigned to the treatment group achieved greater weight loss compared with baseline at the end of the study, on the day of bariatric surgery, and 1 month after surgery, compared with the placebo group (-4. 9 ± 4.9 %, -10.8 ± 5.3 %, and -17.1 ± 3.8 % vs - 0.1 ± 2.6 %, -4.3 ± 3.2 %, and -12.8 ± 4.1 %, respectively, the differences between the groups being significant at all time points (p = 0.015, 0.006, and 0.044, respectively)." does not appear to be confirmed in Fig. 3. Please clarify the results.
7. In agreement with your results the statement in line 166-169 of the discussion, "In our study, which included a sample very similar to the largest clinical trial ever conducted [14], patients in the treatment group showed significantly higher total weight loss, higher percentage of weight loss, as well as higher percentage of fat loss than patients in the control group" is incorrect. Please clarify this point
8. Please report the legend of acronyms in Fig. 2 and Fig. 3
Author Response
Dear Reviewer,
Thank you for giving us the opportunity to improve our article “Endoscopic intragastric injection of botulinum toxin A in obese patients accelerates weight loss after bariatric surgery: follow-up of a randomized controlled trial (IntraTox study).”
The various suggestions have been incorporated into the new version wherever applicable. Please find below our responses and the action taken to all the suggestions and comments.
Please see the attachment to check the new version of the manuscript.
Once again, we very much appreciate all the work with the review.
Yours sincerely,
Dr. Francisco José Sánchez Torralvo
Dr. Gabriel Olveira
I read with interest the study entitled "Endoscopic intragastric injection of botulinum toxin A in obese patients accelerates weight loss after bariatric surgery: follow-up of a randomized controlled trial (IntraTox study)".
The study is interesting but there are some major critical issues that need to be clarified.
How was the sample size calculation done? The authors need to explain this point.
Authors: Thank you for your appreciation. We have included a paragraph on sample size in the material and methods section of the new version of the manuscript.
What type of bariatric surgeries were performed? Please report this information.
A: Thank you for your suggestion. We have included a paragraph on sample size in the results section of the new version of the manuscript.
Was randomization also performed for the different type of surgery?
A: This is a good question. Randomization was performed before the preparatory endoscopy, without knowing the type of bariatric surgery that was going to be performed.
Fig 2 in the result is not easily understood. The weeks counted in the horizontal line are from 2 to 24 and perhaps it could be an error.
A: Thank you for your appreciation. We have improved the figures. The weeks that count from 2 to 24 deal with the preoperative weeks after the endoscopic procedure.
Clarify the number of patients above 50 kg/m2 in the 2 groups.
A: We have provided the information in the new version of the manuscript.
The statement on line 141-146 in the results paragraph: "Only patients with BMI greater than 50 kg/m2 assigned to the treatment group achieved greater weight loss compared with baseline at the end of the study, on the day of bariatric surgery, and 1 month after surgery, compared with the placebo group (-4. 9 ± 4.9 %, -10.8 ± 5.3 %, and -17.1 ± 3.8 % vs - 0.1 ± 2.6 %, -4.3 ± 3.2 %, and -12.8 ± 4.1 %, respectively, the differences between the groups being significant at all time points (p = 0.015, 0.006, and 0.044, respectively)." does not appear to be confirmed in Fig. 3. Please clarify the results.
A: Thank you for your appreciation. We have improved the figures for a better understanding of the statistical significance reached in each week.
In agreement with your results the statement in line 166-169 of the discussion, "In our study, which included a sample very similar to the largest clinical trial ever conducted [14], patients in the treatment group showed significantly higher total weight loss, higher percentage of weight loss, as well as higher percentage of fat loss than patients in the control group" is incorrect. Please clarify this point
A: Thank you for your appreciation. We have redrafted the paragraph as the first statement referred to our previous paper.
8. Please report the legend of acronyms in Fig. 2 and Fig. 3
A: Thank you for your appreciation. We have reported the legend in the new version of the manuscript.
Reviewer 2 Report
In this follow-up study of an RCT, Dr. Sánchez-Torralvo and colleagues continued their observation on the weight reduction effect of botulinum toxin A (BT-A) before, 4 and 12 weeks after bariatric surgery. They concluded that the weight reduction effect is retained until at least 4 weeks after BS and such an effect was seen in patients with BMI > 50 kg/m2. Overall, the manuscript is quite clear and there is potential insight. However, some additional information should be added by the authors and some corrections are required:
- The authors are advised to seek a help from a Native English writer to correct the grammatical errors and typos found all over the manuscript. They sometimes impair the understandability of the sentences. So, this aspect must be improved.
- Since Botox or BT-A works to reduce the contraction / motility of stomach (thereby slowing gastric emptying), the safety component has to be reported in this study as well. Please add the data on the reported adverse effects (minor / major) in each time point (particularly before, 4 weeks after and 3 months after BS).
- "intragastric injection of botulinum toxin A (BT-A) seems effective to achieve weight loss." What is the basis of this statement? For background, the authors need a stronger sentence to support that this study is important to pursue.
- "a follow-up analysis of the IntraTox Study, that randomized 52 patients on bariatric surgery waiting lists in a single-centre, randomised, double-blind, placebo-controlled clinical trial." Grammar needs to be corrected. Apart from it, this sentence is not correct. The authors included 46 subjects, not 52, in this follow-up study. Please revise.
- "The treatment group was administered intragastric BT-A, whereas the control group was administered physiological saline solution." Something that always confuses me when I read the manuscript is the timeline of these interventions. When was the injection of BT-A? How long before the BS? This has to be clarified both in the abstract and the text.
- Please also clarify whether it was a one-time endoscopic injection of BT-A or repeated procedures? Perhaps this should be added in the methods section (Endoscopy).
- "Nowadays, the most effective treatment for obesity option is bariatric surgery, but it entails invasive and has a substantial risk of adverse events compared with other treatments" After this sentence, the authors need to explain the benefits of BS in short and long term, for example in reducing MACE. Also, whether it is superior than lifestyle modification. The authors can check this publication (PMID: 34684569).
- Line 49: "...most of them were case series or clinical trials..."
- Line 52, "botulinum toxin A" should be "BT-A"
- "In our centre, all patients planning to undergo a bariatric surgery have to go through a diagnostic endoscopy several months before the surgery." Is this the endoscopy where BT-A was injected? How many months specifically?
- "to date there are no studies evaluating the effect of intragastric injection of BT-A beyond 6 months of follow-up, as well as its effect after bariatric surgery." Is this what was done in this study? Since the authors did not clearly explain the timeline, we don't know whether this study covers the 6 months after the BT-A injection or not. Please clarify.
- Line 73: "randomised" should be "randomized". Please be consistent between UK spelling and US spelling throughout the manuscript.
- Line 99: "technique" should be "procedure"
- For section 2.2, I would advise the authors to visualize the timeline of observations since at present, it is a bit difficult to understand the time points of this study.
- Line 116 and beyond: Please change "basal visit" to "initial visit"
- In Table 1, when describing the gender percentage (e.g., 85/15) please also add the absolute number in front of it.
- It is also unclear which comorbidities meant by the authors. DM, CVD, HT, dyslipidemia and OSAS are also comorbidities but the numbers of subjects don't add up (maybe because 1 patient could have more than 1 comorbidity?). Please revise and make it clearer.
- Please correct Figure 3. I couldn't find (a) and (b) in the Figure. Something must be missing there, either the BMI <50 or >50.
- Also, please always make sure to add the statistical significant sign (*) in the bar charts. At the moment, it seems that everything was not statistically significant (Figure 2 and 3), but this is not true based on the text.
Author Response
Dear Reviewer,
Thank you for giving us the opportunity to improve our article “Endoscopic intragastric injection of botulinum toxin A in obese patients accelerates weight loss after bariatric surgery: follow-up of a randomized controlled trial (IntraTox study)”
The various suggestions have been incorporated into the new version wherever applicable. Please find below our responses and the action taken to all the suggestions and comments.
Please see the attachment to check the new version of the manuscript.
Once again, we very much appreciate all the work with the review.
Yours sincerely,
Dr. Francisco José Sánchez Torralvo
Dr. Gabriel Olveira
In this follow-up study of an RCT, Dr. Sánchez-Torralvo and colleagues continued their observation on the weight reduction effect of botulinum toxin A (BT-A) before, 4 and 12 weeks after bariatric surgery. They concluded that the weight reduction effect is retained until at least 4 weeks after BS and such an effect was seen in patients with BMI > 50 kg/m2. Overall, the manuscript is quite clear and there is potential insight. However, some additional information should be added by the authors and some corrections are required:
The authors are advised to seek a help from a Native English writer to correct the grammatical errors and typos found all over the manuscript. They sometimes impair the understandability of the sentences. So, this aspect must be improved.
Authors: Thank you for your appreciation. We have carried out an in-depth review of the text, trying to correct the grammatical and spelling errors.
Since Botox or BT-A works to reduce the contraction / motility of stomach (thereby slowing gastric emptying), the safety component has to be reported in this study as well. Please add the data on the reported adverse effects (minor / major) in each time point (particularly before, 4 weeks after and 3 months after BS).
A: We very much appreciate your suggestion. We have added a paragraph on adverse effects.
"intragastric injection of botulinum toxin A (BT-A) seems effective to achieve weight loss." What is the basis of this statement? For background, the authors need a stronger sentence to support that this study is important to pursue.
A: Thank you for your appreciation that improves our manuscript. We have changed the background statement.
"a follow-up analysis of the IntraTox Study, that randomized 52 patients on bariatric surgery waiting lists in a single-centre, randomised, double-blind, placebo-controlled clinical trial." Grammar needs to be corrected. Apart from it, this sentence is not correct. The authors included 46 subjects, not 52, in this follow-up study. Please revise.
A: Thank you for your appreciation. We have revised that statement.
"The treatment group was administered intragastric BT-A, whereas the control group was administered physiological saline solution." Something that always confuses me when I read the manuscript is the timeline of these interventions. When was the injection of BT-A? How long before the BS? This has to be clarified both in the abstract and the text.
A: Thank you for your appreciation. We have tried to improve the understanding of the study timeline with the new wording both in the abstract and the text.
Please also clarify whether it was a one-time endoscopic injection of BT-A or repeated procedures? Perhaps this should be added in the methods section (Endoscopy).
A: It was a one-time endoscopic injection of BT-A. We have clarified it in the new version.
"Nowadays, the most effective treatment for obesity option is bariatric surgery, but it entails invasive and has a substantial risk of adverse events compared with other treatments" After this sentence, the authors need to explain the benefits of BS in short and long term, for example in reducing MACE. Also, whether it is superior than lifestyle modification. The authors can check this publication (PMID: 34684569).
A: We very much appreciate your suggestion. We have added a statement about it.
Line 49: "...most of them were case series or clinical trials..."
A: Thank you for your appreciation. We have made the suggested change.
Line 52, "botulinum toxin A" should be "BT-A"
A: Thank you for your appreciation. We have made the suggested change.
"In our centre, all patients planning to undergo a bariatric surgery have to go through a diagnostic endoscopy several months before the surgery." Is this the endoscopy where BT-A was injected? How many months specifically?
A: We have tried to improve the understanding of the study timeline with the new wording both in the abstract and the text.
"to date there are no studies evaluating the effect of intragastric injection of BT-A beyond 6 months of follow-up, as well as its effect after bariatric surgery." Is this what was done in this study? Since the authors did not clearly explain the timeline, we don't know whether this study covers the 6 months after the BT-A injection or not. Please clarify.
A: As we said previously, we have tried to improve the understanding of the study timeline with the new wording both in the abstract and the text.
Line 73: "randomised" should be "randomized". Please be consistent between UK spelling and US spelling throughout the manuscript.
A: Thank you for your appreciation. We have tried to unify in the US spelling.
Line 99: "technique" should be "procedure"
A: Thank you for your appreciation. We have made the suggested change.
For section 2.2, I would advise the authors to visualize the timeline of observations since at present, it is a bit difficult to understand the time points of this study.
A: We have tried to improve the understanding of the study timeline.
Line 116 and beyond: Please change "basal visit" to "initial visit"
A: Thank you for your appreciation. We have made the suggested changes.
In Table 1, when describing the gender percentage (e.g., 85/15) please also add the absolute number in front of it.
A: Thank you for your suggestion. We have added the absolute number.
It is also unclear which comorbidities meant by the authors. DM, CVD, HT, dyslipidemia and OSAS are also comorbidities but the numbers of subjects don't add up (maybe because 1 patient could have more than 1 comorbidity?). Please revise and make it clearer.
A: Exactly. Some patients had more than one comorbidity. There were no baseline differences regarding comorbidities between groups.
Please correct Figure 3. I couldn't find (a) and (b) in the Figure. Something must be missing there, either the BMI <50 or >50.
A: Thank you for your appreciation. We have improved the figures
Also, please always make sure to add the statistical significant sign (*) in the bar charts. At the moment, it seems that everything was not statistically significant (Figure 2 and 3), but this is not true based on the text.
A: Thank you for your appreciation. We have improved the figures.
Reviewer 3 Report
Interesting article
Talk about some news (besides yours) in bariatric surgery. For example "emerging and innovative field with a clinical need of growing interest for public health".
Cite this review on gastric artery embolization in the introduction (https://doi.org/10.3390/nu13082541)
Tables: no p-values?
Well written article, a little more cautious in the conclusions. Moderate the emphatic tones and always stick to the limitations of the study (small sample size, experimental procedure, confounding bias). Further studies are needed.
Author Response
Dear Reviewer,
Thank you for giving us the opportunity to improve our article “Endoscopic intragastric injection of botulinum toxin A in obese patients accelerates weight loss after bariatric surgery: follow-up of a randomized controlled trial (IntraTox study)”
The various suggestions have been incorporated into the new version wherever applicable. Please find below our responses and the action taken to all the suggestions and comments.
Please see the attachment to check the new version of the manuscript.
Once again, we very much appreciate all the work with the review.
Yours sincerely,
Dr. Francisco José Sánchez Torralvo
Dr. Gabriel Olveira
Interesting article
Talk about some news (besides yours) in bariatric surgery. For example "emerging and innovative field with a clinical need of growing interest for public health".
Cite this review on gastric artery embolization in the introduction (https://doi.org/10.3390/nu13082541)
Authors: We very much appreciate your suggestion that improves our manuscript. We have revised the background and included the proposed reference.
Tables: no p-values?
A: We understand that you refer to the figures. We have added data of statistical significance.
Well written article, a little more cautious in the conclusions. Moderate the emphatic tones and always stick to the limitations of the study (small sample size, experimental procedure, confounding bias). Further studies are needed.
A: We very much appreciate your suggestion. We have changed some statements to be more cautious in the conclusions.
Round 2
Reviewer 2 Report
Thanks for the responses. I have no further comment.